# DATA VALUATION USING REINFORCEMENT LEARNING

## ABSTRACT

Quantifying the value of data is a fundamental problem in machine learning. Data valuation has multiple important use cases: (1) building insights about the learning task, (2) domain adaptation, (3) corrupted sample discovery, and (4) robust learning. To adaptively learn data values jointly with the target task predictor model, we propose a meta learning framework which we name Data Valuation using Reinforcement Learning (DVRL). We employ a data value estimator (modeled by a deep neural network) to learn how likely each datum is used in training of the predictor model. We train the data value estimator using a reinforcement signal of the reward obtained on a small validation set that reflects performance on the target task. We demonstrate that DVRL yields superior data value estimates compared to alternative methods across different types of datasets and in a diverse set of application scenarios. The corrupted sample discovery performance of DVRL is close to optimal in many regimes (i.e. as if the noisy samples were known apriori), and for domain adaptation and robust learning DVRL significantly outperforms state-of-the-art by 14.6% and 10.8%, respectively.

## 1 INTRODUCTION

Data is an essential ingredient in machine learning. Machine learning models are well-known to improve when trained on large-scale and high-quality datasets (Hestness et al., 2017; Najafabadi et al., 2015). However, collecting such large-scale and high-quality datasets is costly and challenging. One needs to determine the samples that are most useful for the target task and then label them correctly. Recent work (Toneva et al., 2019) suggests that not all samples are equally useful for training, particularly in the case of deep neural networks. In some cases, similar or even higher test performance can be obtained by removing a significant portion of training data, i.e. low-quality or noisy data may be harmful (Ferdowsi et al., 2013; Frenay & Verleysen, 2014). There are also some scenarios where train-test mismatch cannot be avoided because the training dataset only exists for a different domain. Different methods (Ngiam et al., 2018; Zhu et al., 2019) have demonstrated the importance of carefully selecting the most relevant samples to minimize this mismatch.

Accurately quantifying the value of data has a great potential for improving model performance for real-world training datasets which commonly contain incorrect labels, and where the input samples differ in relatedness, sample quality, and usefulness for the target task. Instead of treating all data samples equally, lower priority can be assigned for a datum to obtain a higher performance model – for example in the following scenarios:

1. Incorrect label (e.g. human labeling errors).
2. Input comes from a different distribution (e.g. different location or time).
3. Input is noisy or low quality (e.g. noisy capturing hardware).
4. Usefulness for target task (label is very common in the training dataset but not as common in the testing dataset).

In addition to improving performance in such scenarios, data valuation also enables many new use cases. It can suggest better practices for data collection, e.g. what kinds of additional data would the model benefit the most from. For organizations that sell data, it determines the correct value-based pricing of data subsets. Finally, it enables new possibilities for constructing very large-scale training datasets in a much cheaper way, e.g. by searching the Internet using the labels and filtering away less valuable data.

How does one evaluate the value of a single datum? This is a crucial and challenging question. It is straightforward to address at the full dataset granularity: one could naively train a model on the entire dataset and use its prediction performance as the value. However, evaluating the value of each datum is far more difficult, especially for complex models such as deep neural networks that are trained on large-scale datasets. In this paper, we propose a meta learning-based data valuation method which we name Data Valuation using Reinforcement Learning (DVRL). Our method integrates data valuation with the training of the target task predictor model. DVRL determines a reward by quantifying the performance on a small validation set, and uses it as a reinforcement signal to learn the likelihood of each datum being using in training of the predictor model. In a wide range of use cases, including domain adaptation, corrupted sample discovery and robust learning, we demonstrate significant improvements compared to permutation-based strategies (such as Leave-one-out and Influence Function (Koh & Liang, 2017)) and game theory-based strategies (such as Data Shapley (Ghorbani & Zou, 2019)). The main contributions can be summarized as follows:

1. We propose a novel meta learning framework for data valuation that is jointly optimized with the target task predictor model.
2. We demonstrate multiple use cases of the proposed data valuation framework and show DVRL significantly outperforms competing methods on many image, tabular and language datasets.
3. Unlike previous methods, DVRL is scalable to large datasets and complex models, and its computational complexity is not directly dependent on the size of the training set.

## 2 RELATED WORK

**Data valuation:** A commonly-used method for data valuation is leave-one-out (LOO). It quantifies the performance difference when a specific sample is removed and assigns it as that sample's data value. The computational cost is a major concern for LOO – it scales linearly with the number of training samples which means its cost becomes prohibitively high for large-scale datasets and complex models. In addition, there are fundamental limitations in the approximation. For example, if there are two exactly equivalent samples, LOO underestimates the value of that sample even though that sample may be very important. The method of Influence Function (Koh & Liang, 2017) was proposed to approximate LOO in a computationally-efficient manner. It uses the gradient of the loss function with small perturbations to estimate the data value. In order to compute the gradient, Hessian values are needed; however these are prohibitively expensive to compute for neural networks. Approximations for Hessian computations are possible, although they generally result in performance limitations. From data quality assessment perspective, the method of Influence Function inherits the major limitations of LOO.

Data Shapley (Ghorbani & Zou, 2019) is another relevant work. Shapley values are motivated by game theory (Shapley, 1953) and are commonly used in feature attribution problems such as relating predictions to input features (Lundberg & Lee, 2017). For Data Shapley, the prediction performance of all possible subsets is considered and the marginal performance improvement is used as the data value. The computational complexity for computing the exact Shapley value is exponential with the number of samples. Therefore, Monte Carlo sampling and gradient-based estimation are used to approximate them. However, even with these approximations, the computational complexity still remains high (indeed much higher than LOO) due to re-training for each test combination. In addition, the approximations may result in fundamental limitations in data valuation performance – e.g. with Monte Carlo approximation, the ratio of tested combinations compared to all possible combinations decreases exponentially. Moreover, in all the aforementioned methods data valuation is decoupled from predictor model training, which limits the performance due to lack of joint optimization.

**Meta learning-based adaptive learning:** There are relevant studies that utilize meta learning for adaptive weight assignment while training for various use cases such as robust learning, domain adaptation, and corrupted sample discovery. ChoiceNet (Choi et al., 2018) explicitly models output distributions and uses the correlations of the output to improve robustness. Xue et al. (2019) estimates uncertainty of predictions to identify the corrupted samples. Li et al. (2019) combines meta learning with standard stochastic gradient update with generated synthetic noise for robust learning. Shen & Sanghavi (2019) alternates the processes of selecting the samples with low loss and model training to improve robustness. Shu et al. (2019) uses neural networks to model the relationship between current loss and the corresponding sample weights, and utilizes a meta-learning

framework for robust weight assignment. Köhler et al. (2019) estimates the uncertainty to discover the noisy labeled data and relabels mislabeled samples to improve the prediction performance of the predictor model. Gold Loss Correction (Hendrycks et al., 2018) uses a clean validation set to recover the label corruption matrix to re-train the predictor model with corrected labels. Learning to Reweight (Ren et al., 2018) proposes a single gradient descent step guided with validation set performance to reweight the training batch. Domain Adaptive Transfer Learning (Ngiam et al., 2018) introduces importance weights (based on the prior label distribution match) to scale the training samples for transfer learning. MentorNet (Jiang et al., 2018) proposes a curriculum learning framework that learns the order of mini-batch for training of the corresponding predictor model. Our method, DVRL, differs from the aforementioned as it directly models the value of the data using learnable neural networks (which we refer to as a data value estimator). To train the data value estimator, we use reinforcement learning with a sampling process. DVRL is model-agnostic and even applicable to non-differentiable target objectives. Learning is jointly performed for the data value estimator and the corresponding predictor model, yielding superior results in all of the use cases we consider.

# 3 PROPOSED METHOD

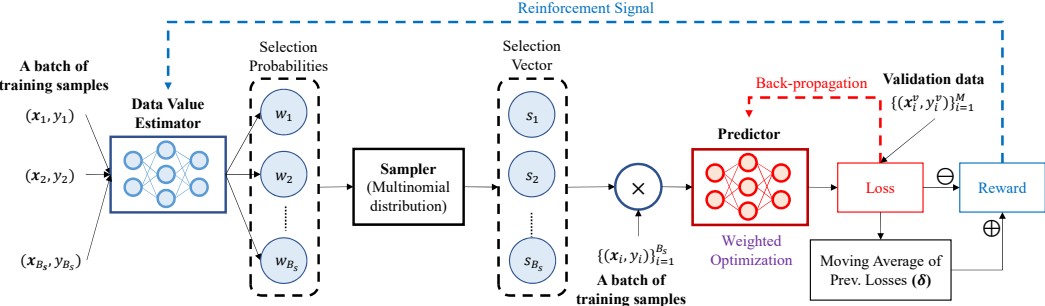

Figure 1: Block diagram of the DVRL framework for training. A batch of training samples is used as the input to the data value estimator (with shared parameters across the batch) and the output corresponds to selection probabilities: $w_i = h_\phi(\mathbf{x}_i, y_i)$ of a multinomial distribution. The sampler, based on this multinomial distribution, returns the selection vector $\mathbf{s} = (s_1, ..., s_{B_s})$ where $s_i \in \{0, 1\}$ and $P(s_i = 1) = w_i$. The target task predictor model is trained only using the samples with selection vector $s_i = 1$, using conventional gradient-descent optimization. The selection probabilities $w_i$ rank the samples according to their importance – these are used as data values. The loss of the predictor model is evaluated on a small validation set, which is compared to the moving average of the previous losses ($\delta$) to determine the reward. Finally, the reinforcement signal guided by this reward updates the data value estimator. Block diagrams for inference are shown in Appendix A.

**Framework:** Let us denote the training dataset as $\mathcal{D} = \{(\mathbf{x}_i, y_i)\}_{i=1}^N \sim \mathcal{P}$ where $\mathbf{x}_i \in \mathcal{X}$ is a feature vector in the $d$-dimensional feature space $\mathcal{X}$, e.g. $\mathbb{R}^d$, and $y_i \in \mathcal{Y}$ is a corresponding label in the label space $\mathcal{Y}$, e.g. $\Delta[0, 1]^c$ for classification where $c$ is the number of classes and $\Delta$ is the simplex. We consider a disjoint testing dataset $\mathcal{D}^t = \{(\mathbf{x}_j^t, y_j^t)\}_{j=1}^M \sim \mathcal{P}^t$ where the target distribution $\mathcal{P}^t$ does not need to be the same with the training distribution $\mathcal{P}$. We assume an availability of a (often small[1]) validation dataset $\mathcal{D}^v = \{(\mathbf{x}_k^v, y_k^v)\}_{k=1}^L \sim \mathcal{P}^t$ that comes from the target distribution $\mathcal{P}^t$.

The DVRL method (overview in Fig. 1) consists of two learnable functions: (1) the target task predictor model $f_\theta$, (2) data value estimator model $h_\phi$. The predictor model $f_\theta : \mathcal{X} \to \mathcal{Y}$ is trained to minimize a certain weighted loss function $\mathcal{L}_f$ (e.g. Mean Squared Error (MSE) for regression or cross entropy for classification) on training set $\mathcal{D}$:

$$f_\theta = \arg\min_{\hat{f} \in \mathcal{F}} \frac{1}{N} \sum_{i=1}^N h_\phi(\mathbf{x}_i, y_i) \cdot \mathcal{L}_f(\hat{f}(\mathbf{x}_i), y_i). \tag{1}$$

$f_\theta$ can be any trainable function with parameters $\theta$, such as a neural network. The data value estimator model $h_\phi : \mathcal{X} \cdot \mathcal{Y} \to [0, 1]$, on the other hand, is optimized to output weights that determine

---

[1]We provide empirical results that how small the validation set can be in Section 4.5.

the distribution of selection likelihood of the samples to train the predictor model $f_\theta$. We formulate the corresponding optimization problem as:

$$\min_{h_\phi} \quad \mathbb{E}_{(\mathbf{x}^v, y^v) \sim P^t} \Big[ \mathcal{L}_h(f_\theta(\mathbf{x}^v), y^v) \Big]$$
$$\text{s.t.} \quad f_\theta = \arg\min_{\hat{f} \in \mathcal{F}} \mathbb{E}_{(\mathbf{x}, y) \sim P} \Big[ h_\phi(\mathbf{x}, y) \cdot \mathcal{L}_f(\hat{f}(\mathbf{x}), y) \Big] \tag{2}$$

where $h_\phi(\mathbf{x}, y)$ represents value of the training sample $(\mathbf{x}, y)$. The data value estimator is also a trainable function, such as a neural network. Similar to $\mathcal{L}_f$, we use MSE or cross entropy for $\mathcal{L}_h$.

**Training:** To encourage exploration based on uncertainty, we model training sample selection stochastically. Let $w = h_\phi(\mathbf{x}, y)$ denote the probability that $(\mathbf{x}, y)$ is used to train the predictor model $f_\theta$. $h_\phi(\mathcal{D}) = \{h_\phi(\mathbf{x}_i, y_i)\}_{i=1}^N$ is the probability distribution for inclusion of each training sample. $\mathbf{s} \in \{0, 1\}^N$ is a binary vector that represents the selected samples. If $s_i = 1/0$, $(\mathbf{x}_i, y_i)$ is selected/not selected for training the predictor model. $\pi_\phi(\mathcal{D}, \mathbf{s}) = \prod_{i=1}^N \big[ h_\phi(\mathbf{x}_i, y_i)^{s_i} \cdot (1 - h_\phi(\mathbf{x}_i, y_i))^{1-s_i} \big]$ is the probability that certain selection vector $\mathbf{s}$ is selected based on $h_\phi(\mathcal{D})$. We assign the outputs of the data value estimator model, $w = h_\phi(\mathbf{x}, y)$, as the data values. We can use the data values to rank the dataset samples (e.g. to determine a subset of the training dataset) and to do sample-adaptive training (e.g. for domain adaptation).

The predictor model can be trained using standard stochastic gradient descent because it is differentiable with respect to the input. However, gradient descent-based optimization cannot be used for the data value estimator because the sampling process is non-differentiable. There are multiple ways to handle the non-differentiable optimization bottleneck, such as Gumbel-softmax (Jang et al., 2017) or stochastic back-propagation (Rezende et al., 2014). In this paper, we consider reinforcement learning instead, which directly encourages exploration of the policy towards the optimal solution of Eq. (2). We use the REINFORCE algorithm (Williams, 1992) to optimize the policy gradients, with the rewards obtained from a small validation set that approximates performance on the target task. For the loss function $\hat{l}(\phi)$:

$$\hat{l}(\phi) = \mathbb{E}_{(\mathbf{x}^v, y^v) \sim P^t} \Big[ \mathbb{E}_{\mathbf{s} \sim \pi_\phi(\mathcal{D}, \cdot)} \big[ \mathcal{L}_h(f_\theta(\mathbf{x}^v), y^v) \big] \Big]$$
$$= \int P^t(\mathbf{x}^v) \Big[ \sum_{\mathbf{s} \in [0,1]^N} \pi_\phi(\mathcal{D}, \mathbf{s}) \cdot \big[ \mathcal{L}_h(f_\theta(\mathbf{x}^v), y^v) \big] \Big] d\mathbf{x}^v,$$

we directly compute the gradient $\nabla_\phi \hat{l}(\phi)$ as:

$$\nabla_\phi \hat{l}(\phi) = \int P^t(\mathbf{x}^v) \Big[ \sum_{\mathbf{s} \in [0,1]^N} \nabla_\phi \pi_\phi(\mathcal{D}, \mathbf{s}) \cdot \big[ \mathcal{L}_h(f_\theta(\mathbf{x}^v), y^v) \big] \Big] d\mathbf{x}^v$$
$$= \int P^t(\mathbf{x}^v) \Big[ \sum_{\mathbf{s} \in [0,1]^N} \nabla_\phi \log(\pi_\phi(\mathcal{D}, \mathbf{s})) \cdot \pi_\phi(\mathcal{D}, \mathbf{s}) \cdot \big[ \mathcal{L}_h(f_\theta(\mathbf{x}^v), y^v) \big] \Big] d\mathbf{x}^v$$
$$= \mathbb{E}_{(\mathbf{x}^v, y^v) \sim P^t} \Big[ \mathbb{E}_{\mathbf{s} \sim \pi_\phi(\mathcal{D}, \cdot)} \big[ \mathcal{L}_h(f_\theta(\mathbf{x}^v), y^v) \big] \nabla_\phi \log(\pi_\phi(\mathcal{D}, \mathbf{s})) \Big],$$

where $\nabla_\phi \log(\pi_\phi(\mathcal{D}, \mathbf{s}))$ is

$$\nabla_\phi \log(\pi_\phi(\mathcal{D}, \mathbf{s})) = \nabla_\phi \sum_{i=1}^N \log \Big[ h_\phi(\mathbf{x}_i, y_i)^{s_i} \cdot (1 - h_\phi(\mathbf{x}_i, y_i))^{1-s_i} \Big]$$
$$= \sum_{i=1}^N s_i \nabla_\phi \log \big[ h_\phi(\mathbf{x}_i, y_i) \big] + (1 - s_i) \nabla_\phi \log \big[ (1 - h_\phi(\mathbf{x}_i, y_i)) \big].$$

To improve the stability of the training, we use the moving average of the previous loss ($\delta$), with a window size ($T$), as the baseline for the current loss. The pseudo-code is shown in Algorithm 1.

**Computational complexity:** DVRL models the mapping between an input and its value with a learnable function. The training time of DVRL is not directly proportional to the dataset size, but rather dominated by the required number of iterations and per-iteration complexity in Algorithm 1. One way to minimize the computational overhead is to use pre-trained models to initialize the

---

**Algorithm 1** Pseudo-code of DVRL training

---

1: **Inputs:** Learning rates $\alpha, \beta > 0$, mini-batch size $B_p, B_s > 0$, inner iteration count $N_I > 0$, moving average window $T > 0$, training dataset $\mathcal{D}$, validation dataset $\mathcal{D}^v = \{(\mathbf{x}_k^v, y_k^v)\}_{k=1}^L$

2: **Initialize** parameters $\theta, \phi$, moving average $\delta = 0$

3: **while** until convergence **do**

4:      Sample a mini-batch from the entire training dataset: $\mathcal{D}_B = (\mathbf{x}_j, y_j)_{j=1}^{B_s} \sim \mathcal{D}$

5:      **for** $j = 1, ..., B_s$ **do**

6:          Calculate selection probabilities: $w_j = h_\phi(\mathbf{x}_j, y_j)$

7:          Sample selection vector: $s_j \sim Ber(w_j)$

8:      **for** $t = 1, ..., N_I$ **do**

9:          Sample a mini-batch $(\tilde{\mathbf{x}}_m, \tilde{y}_m, \tilde{s}_m)_{m=1}^{B_p} \sim (\mathbf{x}_j, y_j, s_j)_{j=1}^{B_s}$

10:          Update the predictor model network parameters $\theta$

$$\theta \leftarrow \theta - \alpha \frac{1}{B_p} \sum_{m=1}^{B_p} \tilde{s}_m \cdot \nabla_\theta \mathcal{L}_f(f_\theta(\tilde{\mathbf{x}}_m), \tilde{y}_m))$$

11:      Update the data value estimator model network parameters $\phi$

$$\phi \leftarrow \phi - \beta \Big[ \frac{1}{L} \sum_{k=1}^L [\mathcal{L}_h(f_\theta(\mathbf{x}_k^v), y_k^v)] - \delta \Big] \nabla_\phi \log \pi_\phi(\mathcal{D}_B, (s_1, ..., s_{B_s}))$$

12:      Update the moving average baseline ($\delta$): $\delta \leftarrow \frac{T-1}{T}\delta + \frac{1}{LT} \sum_{k=1}^L [\mathcal{L}_h(f_\theta(\mathbf{x}_k^v), y_k^v)]$

---

predictor networks at each iteration. Unlike alternative methods like Data Shapley, we demonstrate the scalability of DVRL to large-scale datasets such as CIFAR-100, and complex models such as ResNet-32 (He et al., 2016) and WideResNet-28-10 (Zagoruyko & Komodakis, 2016). Instead of being exponential in terms of the dataset size, the training time overhead DVRL is only twice of conventional training. Please see Appendix D for further analysis on learning dynamics of DVRL and Appendix B for additional computational complexity discussions.

## 4 EXPERIMENTS

We evaluate data value estimation quality of DVRL on multiple types of dataset and use cases.

**Benchmark methods:** We consider the following benchmarks: (1) Randomly-assigned values (Random), (2) Leave-one-out (LOO), (3) Data Shapley Value (Data Shapley) (Ghorbani & Zou, 2019). For some experiments, we also compare with (4) Learning to Reweight (Ren et al., 2018), (5) MentorNet (Jiang et al., 2018), and (6) Influence Function (Koh & Liang, 2017).

**Datasets:** We consider 12 public datasets (3 public tabular datasets, 7 public image datasets, and 2 public language datasets) to evaluate DVRL in comparison to multiple benchmark methods. 3 public tabular datasets are (1) Blog, (2) Adult, (3) Rossmann; 7 public image datasets are (4) HAM 10000, (5) MNIST, (6) USPS, (7) Flower, (8) Fashion-MNIST, (9) CIFAR-10, (10) CIFAR-100; 2 public language datasets are (11) Email Spam, (12) SMS Spam. Details can be found in the hyper-links.

**Baseline predictor models:** We consider various machine learning models as the baseline predictor model to highlight the proposed *model-agnostic* data valuation framework. For Adult and Blog datasets, we use LightGBM (Ke et al., 2017), and for Rossmann dataset, we use XGBoost and multi-layer perceptrons due to their superior performance on the tabular datasets. For Flower, HAM 10000, and CIFAR-10 datasets, we use Inception-v3 with top-layer fine-tuning (pre-trained on ImageNet, (Szegedy et al., 2016)) as the baseline predictor model. For Fashion-MNIST, MNIST, and USPS datasets, we use multinomial logistic regression, and for Email and SMS datasets, we use Naive Bayes model. We also use ResNet-32 (He et al., 2016) and WideResNet-28-10 (Zagoruyko & Komodakis, 2016) as the baseline models for CIFAR-10 and CIFAR-100 datasets in Section 4.3 to demonstrate the scalability of DVRL. For data value estimation network, we use multi-layer perceptrons with ReLU activation as the base architecture. The number of layers and hidden units are optimized with cross-validation.

**Experimental details:** In all experiments, we use Standard Normalizer to normalize the entire features to have zero mean and one standard deviation. We transform categorical variables into one-hot encoded embeddings. We set the inner iteration count ($N_I$=200) for the predictor network, moving average window ($T$=20), and mini-batch size ($B_p$=256) for the predictor network and mini-batch size ($B_s$=2000) for the DVE network (large batch size often improves the stability of the reinforcement learning model training (McCandlish et al., 2018)). We set the learning rate to 0.01 ($\beta$) for the data value estimator (DVE) and 0.001 ($\alpha$) for the predictor network. As the DVE architecture, for tabular datasets, we use 5-layer perceptrons with 100 hidden units and ReLU; and for image datasets, we use 5-layer perceptrons with 100 hidden units and ReLU on top of the CNN-based architecture used for the predictor network (such as ResNet-32 or WideResNet-28-10 in Table 1). In order to provide further informative signal to DVE, we propose to use an additional input of the difference between the predictions of a separate predictive model (fined-tuned or trained from scratch on the validation set) for the training samples and the original training labels. We simply concatenate this additional input to the hidden states of DVE network. Intuitively, if the training label is corrupted, the additional input would be high; thus, this could be an important signal for DVE to assign low value to this sample. Ablation study for the variants of DVRL can be found in the Appendix C.6.

### 4.1 REMOVING HIGH/LOW VALUE SAMPLES

Removing low value samples from the training dataset can improve the predictor model performance, especially in the cases where the training dataset contains corrupted samples. On the other hand, removing high value samples, especially if the dataset is small, would decrease the performance significantly. Overall, the performance after removing high/low value samples is a strong indicator for the quality of data valuation.

Initially, we consider the conventional supervised learning setting, where all training, validation and testing datasets come from the same distribution (without sample corruption or domain mismatch). We use two tabular datasets (Adult and Blog) with 1,000 training samples and one image dataset (Flower) with 2,000 training samples.[2] We use 400 validation samples for tabular datasets and 800 validation samples for the image dataset. Then, we report the prediction performance on the disjoint testing set after removing the high/low value samples based on the estimated data values.

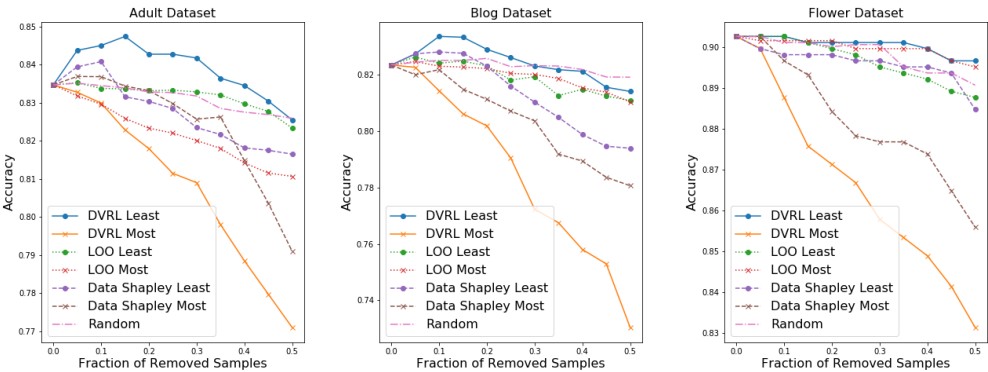

Figure 2: Performance after removing the most (marked with ×) and least (marked with ◯) important samples according to the estimated data values in a conventional supervised learning setting.

As shown in Fig. 2, even in the absence of sample corruption or domain mismatch, DVRL can marginally improve the prediction performance after removing some portion of the least important samples. Using only ∼60%-70% of the training set (the highest valued samples), DVRL can obtain a similar performance compared to training on the entire dataset. After removing a small portion (10%-20%) of the most important samples, the prediction performance significantly degrades which indicates the importance of the high valued samples. Qualitatively looking at these samples, we ob-

---

[2]We use the small training dataset in this experiment in order to compare with LOO and Data Shapley which have high computational complexities. DVRL is scalable to larger datasets as shown in Section 4.3.

serve them to typically be representative of the target task which can be insightful. Overall, DVRL shows the fastest performance degradation after removing the most important samples and the slowest performance degradation after removing the least important samples in most cases, underlining the superiority of DVRL in data valuation quality compared to competing methods.

Next, we focus on the setting of removing high/low value samples in the presence of label noise in the training data. We consider three image datasets: Fashion-MNIST, HAM 10000, and CIFAR-10. As noisy samples hurt the performance of the predictor model, an optimal data value estimator with a clean validation dataset should assign lowest values to the noisy samples. With the removal of samples with noisy labels ('Least' setting), the performance should either increase, or at least decrease much slower, compared to removal of samples with correct labels ('Most' setting). In this experiment, we introduce label noise to 20% of the samples by replacing true labels with random labels. As can be seen in Fig. 10, for all data valuation methods the prediction performance tends to first slowly increase and then decrease in the 'Least' setting; and tends to rapidly decrease in the 'Most' setting. Yet, DVRL achieves the slowest performance decrease in 'Least' setting and fastest performance decrease in the 'Most' setting, reflecting its superiority in data valuation.

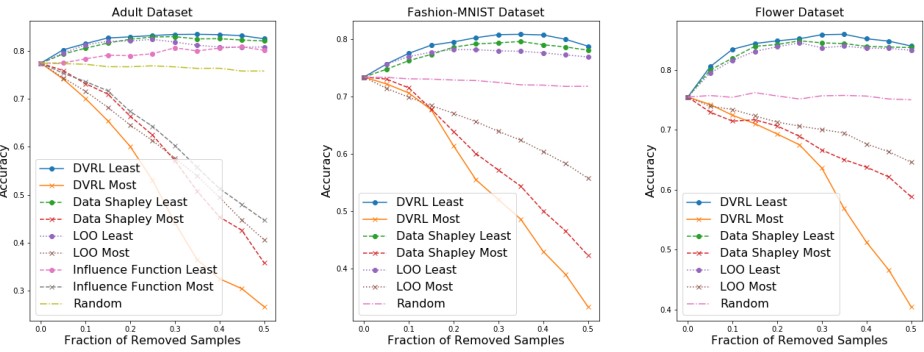

Figure 3: Prediction performance after removing the most (marked with ×) and least (marked with ◯) important samples according to the estimated data values with 20% noisy label ratio. Additional results on Blog, HAM 10000, and CIFAR-10 datasets can be found in Appendix C.3. The prediction performance is lower than state of the art due to a smaller training set size and the introduced noise.

## 4.2 Corrupted sample discovery

There are some scenarios where training samples may contain corrupted samples, e.g. due to cheap label collection methods. An automated corrupted sample discovery method would be highly beneficial for distinguishing samples with clean vs. noisy labels. Data valuation can be used in this setting by having a small clean validation set to assign low data values to the potential samples with noisy labels. With an optimal data value estimator, all noisy labels would get the lowest data values.

We consider the same experimental setting with the previous subsection with 20% noisy label ratio on 6 datasets. Fig. 4 shows that DVRL consistently outperforms all benchmarks (Data Shapley, LOO and Influence Function). The trend of noisy label discovery for DVRL can be very close to optimal (as if we perfectly knew which samples have noisy labels), particularly for the Adult, CIFAR-10 and Flower datasets. To highlight the stability of DVRL, we provide the confidence intervals of DVRL performance on the corrupted sample discovery in Appendix E.

## 4.3 Robust learning with noisy labels

In this section, we consider how reliably DVRL can learn with noisy data in an end-to-end way, without removing the low-value samples as in the previous section. Ideally, noisy samples should get low data values as DVRL converges and a high performance model can be returned. To compare DVRL with two recently-proposed benchmarks: MentorNet (Jiang et al., 2018) and Learning to Reweight (Ren et al., 2018) for this use case, we focus on two complex deep neural networks as the baseline predictor models, ResNet-32 (He et al., 2016) and WideResNet-28-10 (Zagoruyko &

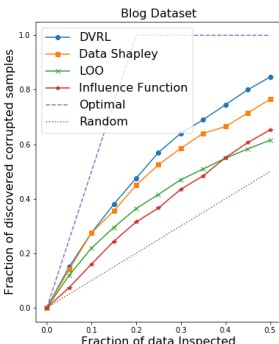 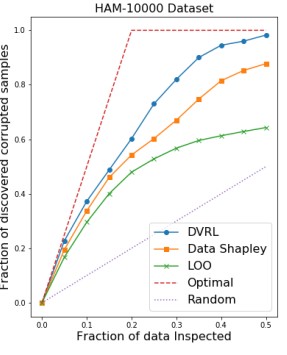 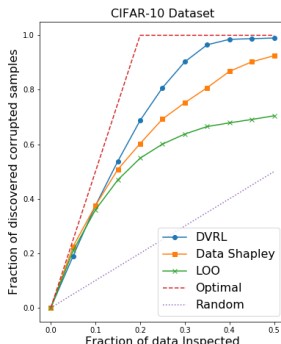

Figure 4: Discovering corrupted samples in three datasets with 20% noisy label ratio. 'Optimal' saturates at 20%, perfectly assigning the lowest data value scores to the samples with noisy labels. 'Random' does not introduce any knowledge on distinguishing clean vs. noisy labels, and thus the fraction of discovered corrupt samples is proportional to the amount of inspection. More results on Adult, Fashion-MNIST and Flower datasets are in Appendix C.4.

Komodakis, 2016), trained on CIFAR-10 and CIFAR-100 datasets. Additional results on other image datasets are in Appendix C.1, and results on robust learning with noisy features are in Appendix C.2.

We consider the same experimental setting from Ren et al. (2018) on CIFAR-10 and CIFAR-100 datasets. For the first experiment, we use WideResNet-28-10 as the baseline predictor model and apply 40% of label noise uniformly across all classes. We use 1,000 clean (noise-free) samples as the validation set and test the performance on the clean testing set. For the second experiment, we use ResNet-32 as the baseline predictor model and apply 40% background noise (same-class noise to the 40% of the samples). In this case, we only use 10 clean samples per class as the validation set. We consider five additional benchmarks: (1) *Validation Set Only* – which only uses clean validation set for training, (2) *Baseline* – which only uses noisy training set for training, (3) *Baseline + Fine-tuning* – which is initialized with the trained baseline model on the noisy training set and fine-tuned on the clean validation set, (4) *Clean Only (60% data)* – which is trained on the clean training set after removing the training samples with flipped labels, (5) *Zero Noise* – which uses the original noise-free training set for training (100% clean training data). We exclude Data Shapley and LOO in this experiment due to their prohibitively-high computational complexities.

| Noise (predictor model) | **Uniform** (WideResNet-28-10) | | **Background** (ResNet-32) | |
|---|---|---|---|---|
| Datasets | CIFAR-10 | CIFAR-100 | CIFAR-10 | CIFAR-100 |
| Validation Set Only | $46.64 \pm 3.90$ | $9.94 \pm 0.82$ | $15.90 \pm 3.32$ | $8.06 \pm 0.76$ |
| Baseline | $67.97 \pm 0.62$ | $50.66 \pm 0.24$ | $59.54 \pm 2.16$ | $37.82 \pm 0.69$ |
| Baseline + Fine-tuning | $78.66 \pm 0.44$ | $54.52 \pm 0.40$ | $82.82 \pm 0.93$ | $54.23 \pm 1.75$ |
| MentorNet + Fine-tuning | $78.00$ | $59.00$ | - | - |
| Learning to Reweight | $86.92 \pm 0.19$ | $61.34 \pm 2.06$ | $86.73 \pm 0.48$ | $59.30 \pm 0.60$ |
| **DVRL** | $\mathbf{89.02 \pm 0.27}$ | $\mathbf{66.56 \pm 1.27}$ | $\mathbf{88.07 \pm 0.35}$ | $\mathbf{60.77 \pm 0.57}$ |
| Clean Only (60% Data) | $94.08 \pm 0.23$ | $74.55 \pm 0.53$ | $90.66 \pm 0.27$ | $63.50 \pm 0.33$ |
| Zero Noise | $95.78 \pm 0.21$ | $78.32 \pm 0.45$ | $92.68 \pm 0.22$ | $68.12 \pm 0.21$ |

Table 1: Robust learning with noisy labels. Test accuracy for ResNet-32 and WideResNet-28-10 on CIFAR-10 and CIFAR-100 datasets with 40% of Uniform and Background noise on labels.

As shown in Table 1, DVRL outperforms other robust learning methods in all cases. The performance improvements with DVRL are larger with Uniform noise. Learning to Reweight loses 7.16% and 13.21% accuracy compared to the optimal case (Zero Noise); on the other hand, DVRL only loses 5.06% and 7.99% accuracy for CIFAR-10 and CIFAR-100 respectively with Uniform noise.

## 4.4 Domain adaptation

In this section, we consider the scenario where the training dataset comes from a substantially different distribution from the validation and testing sets. Naive training methods (i.e. equal treatment of all training samples) often fail in this scenario (Ganin et al., 2016; Glorot et al., 2011). Data valuation is expected to be beneficial for this task by selecting the samples from the training dataset that best match the distribution of the validation dataset.

| Source | Target | Task | *Baseline* | Data Shapley | **DVRL** |
|---|---|---|---|---|---|
| Google | HAM10000 | Skin Lesion Classification | .296 | .378 | **.448** |
| MNIST | USPS | Digit Recognition | .308 | .391 | **.472** |
| Email | SMS | Spam Detection | .684 | .864 | **.903** |

Table 2: Domain adaptation setting showing target accuracy. *Baseline* represents the predictor model which is naively trained on the training set with equal treatment of all training samples.

We initially focus on the three cases from Ghorbani & Zou (2019), shown in Table 2. (1) uses Google image search results (cheaply collected dataset) to predict skin lesion classification on HAM 10000 data (clean), (2) uses MNIST data to recognize digit on USPS dataset, (3) uses Email spam data to detect spam in an SMS dataset. The experimental settings are exactly the same with Ghorbani & Zou (2019). Table 2 shows that DVRL significantly outperforms *Baseline* and Data Shapley in all three tasks. One primary reason is that DVRL jointly optimizes the data value estimator and corresponding predictor model; on the other hand, Data Shapley needs a two step processes to construct the predictor model in domain adaptation setting.

Next, we focus on a real-world tabular data learning problem where the domain differences are significant. We consider the sales forecasting problem with the Rossmann Store Sales dataset, which consists of sales data from four different store types. Simple statistical investigation shows a significant discrepancy between the input feature distributions across different store types, meaning there is a large domain mismatch across store types (see Appendix F). To further illustrate distribution difference across the store types, we show the t-SNE analysis on the final layer of a discriminative neural network trained on the entire dataset in Appendix Fig. 11. We consider three different settings: (1) training on all store types (*Train on All*), (2) training on store types excluding the store type of interest (*Train on Rest*), and (3) training only on the store type of interest (*Train on Specific*). In all cases, we evaluate the performance on each store type separately. For example, to evaluate the performance on store type D, *Train on All* setting uses all four store type datasets for training, *Train on Rest* setting uses store types A, B and C for training, and *Train on Specific* setting only uses the store type D for training. *Train on Rest* is expected to yield the largest domain mismatch between training and testing sets, and *Train on Specific* yield the minimal.

| Predictor Model | Store | *Train on All* | | *Train on Rest* | | *Train on Specific* | |
|---|---|---|---|---|---|---|---|
| (Metric: RMSPE) | Type | *Baseline* | DVRL | *Baseline* | DVRL | *Baseline* | DVRL |
| XGBoost | A | 0.1736 | **0.1594** | 0.2369 | **0.2109** | 0.1454 | **0.1430** |
| | B | 0.1996 | **0.1422** | 0.7716 | **0.3607** | 0.0880 | **0.0824** |
| | C | 0.1839 | **0.1502** | 0.2083 | **0.1551** | 0.1186 | **0.1170** |
| | D | 0.1504 | **0.1441** | 0.1922 | **0.1535** | 0.1349 | **0.1221** |
| Neural Networks | A | 0.1531 | **0.1428** | 0.3124 | **0.2014** | 0.1181 | **0.1066** |
| | B | 0.1529 | **0.0979** | 0.8072 | **0.5461** | 0.0683 | **0.0682** |
| | C | 0.1620 | **0.1437** | 0.2153 | **0.1804** | 0.0682 | **0.0677** |
| | D | 0.1459 | **0.1295** | 0.2625 | **0.1624** | 0.0759 | **0.0708** |

Table 3: Performance of *Baseline* and DVRL in 3 different settings with 2 different predictor models on the Rossmann Store Sales dataset. Metric is Root Mean Squared Percentage Error (RMSPE, lower the better). We use 79% of the data as training, 1% as validation, and 20% as testing. DVRL outperforms *Baseline* in all settings.

We evaluate the performance of *Baseline* (train the predictor model without data valuation) and DVRL in 3 different settings with 2 different predictor models (XGBoost (Chen & Guestrin, 2016) and Neural Networks (3-layer perceptrons)). As shown in Table 3, DVRL improves the performance in all settings. The improvements are most significant in *Train on Rest* setting due to the large domain mismatch. For instance, DVRL reduces the error more than 50% for store type B predictions with XGBoost in comparison to *Baseline*. In *Train on All* setting, the performance improvement is still significant, showing that DVRL can distinguish the samples from the target distribution. In Appendix G, we demonstrate that DVRL actually prioritizes selection of the samples from the target store type. In *Train on Specific* setting, the performance improvements are smaller – even without domain mismatch, DVRL can marginally improve the performance by accurately prioritizing the important samples within the same store type. These results further support the conclusions from Fig. 2 in the conventional supervised learning setting that DVRL learns high quality data value scores. Comparison to other domain adaptation benchmarks can be found in Appendix C.5.

### 4.5 Discussion: How many validation samples are needed?

DVRL requires a validation dataset from the target distribution that the testing dataset comes from. Depending on the task, the requirements for the validation dataset may involve being noise-free in labels, being from the same domain, or being high quality. Acquiring such a dataset can be costly in some scenarios and it is desirable to minimize its size requirements.

We analyze the impact of the size of the validation dataset on DVRL with 3 different datasets: Adult, Blog, and Fashion MNIST for the use case of corrupted sample discovery. Similar to Section 4.2, we add 20% noise to the training samples and try to find the corrupted samples with DVRL. As shown in Fig. 5, DVRL achieves reasonable performance with 100 to 400 validation samples. In the Adult dataset, even 10 validation samples are sufficient to achieve a reasonable data valuation quality. Both of these settings are often realistic in real world scenarios.

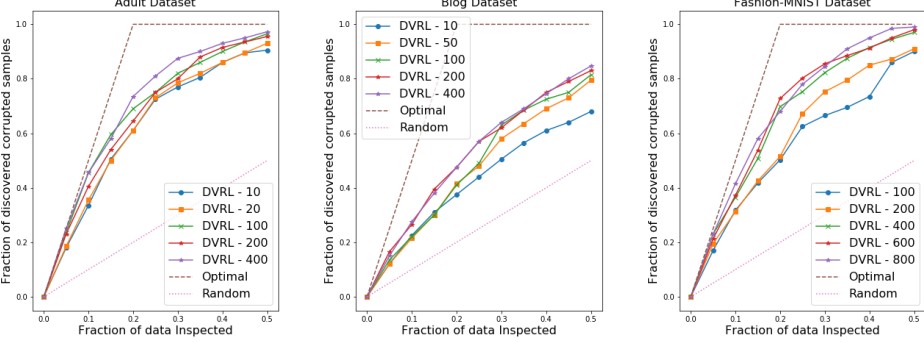

Figure 5: Number of validation samples needed for DVRL. Discovering corrupted samples in three datasets (Adult, Blog and Fashion MNIST) with various number of validation samples. X-axis represents the fraction of inspected data and y-axis is the fraction of discovered corrupted samples. On Adult and Fashion-MNIST datasets, DVRL needs 13% and 14% of inspected samples to identify 50% of the corrupted samples respectively - merely 3% and 4% more than the optimal cases.

## 5 Conclusions

In this paper, we propose a meta learning framework, named DVRL, that adaptively learns data values jointly with a target task predictor model. The value of each datum determines how likely it will be used in training of the predictor model. We model this data value estimation task using a deep neural network, which is trained using reinforcement learning with a reward obtained from a small validation set that represents the target task performance. With a small validation set, DVRL can provide computationally highly efficient and high quality ranking of data values for the training dataset that is useful for domain adaptation, corrupted sample discovery and robust learning. We show that DVRL significantly outperforms other techniques for data valuation in various applications on diverse types of datasets.

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

# A    BLOCK DIAGRAMS FOR INFERENCE

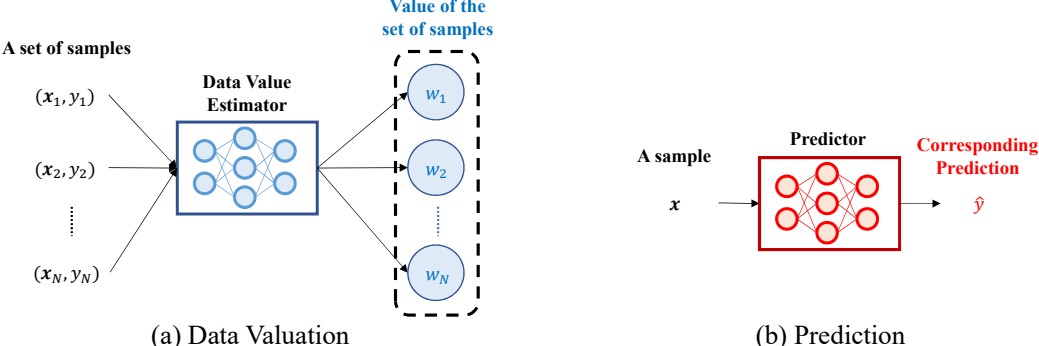

(a) Data Valuation            (b) Prediction

Figure 6: Block diagram of the proposed DVRL framework at inference time. (a) Data valuation, (b) Prediction. For data valuation, the input is a set of samples and the outputs are the corresponding data values. For prediction, the input is a sample and the output is the corresponding prediction. Both the data value estimator and predictor are fixed (not trained) at inference time.

# B    COMPUTATIONAL COMPLEXITY

DVRL first trains the baseline model using the entire dataset (without re-weighting). Afterwards, we can use this pre-trained baseline model to initialize the predictor network and apply fine-tuning with DVRL update steps. The convergence of the fine-tuning process is much faster than the convergence of training from the scratch.

We quantify the computational overhead of DVRL on the CIFAR-100 dataset (consisting 50k training samples) with ResNet-32 as a representative example. Overall, DVRL training takes less than 8 hours (given a pre-trained ResNet-32 model on the entire dataset) on a single NVIDIA Tesla V100 GPU without any hardware optimizations. The pre-training time of ResNet-32 on the entire dataset (without re-weighting) is less than 4 hours; thus the total training time of DVRL is less than 12 hours from the scratch. On the other hand, the training time of Data Shapley (the most competitive benchmark) is more than a week on Fashion MNIST (consisting lower dimensional inputs and less number of classes) with a much simpler predictor model architecture (2-layered convolutional neural networks).

At inference, the data value estimator can be used to obtain data value for each sample. The runtime of data valuation is typically much faster (less than 1 ms per sample) than the predictor model (e.g. ResNet-32 model).

## C    ADDITIONAL RESULTS

### C.1    ADDITIONAL RESULTS ON ROBUST LEARNING WITH NOISY LABELS

We evaluate how DVRL can provide robustness for learning with noisy labels. We add various levels of label noise, ranging from 0% to 50%, to the training sets and evaluate how robust the proposed model (DVRL) is for the noisy dataset. In this experiment, we use three image datasets (CIFAR-10, Flower, and HAM 10000). Note that we initialize the predictor model using pre-trained Inception-v3 networks on ImageNet and only fine-tune the top layer (transfer learning setting).

| Noise | CIFAR-10 | | | Flower | | | HAM 10000 | | |
|---|---|---|---|---|---|---|---|---|---|
| ratio | *Clean* | DVRL | *Baseline* | *Clean* | DVRL | *Baseline* | *Clean* | DVRL | *Baseline* |
| 0% | .8297 | **.8305** | .8297 | .9090 | **.9292** | .9090 | .7129 | **.7148** | .7129 |
| 10% | .8281 | **.8306** | .7713 | .9057 | **.9158** | .7441 | .7094 | **.7142** | .6746 |
| 20% | **.8285** | .8271 | .6883 | .9026 | **.9152** | .5960 | .7098 | **.7126** | .6199 |
| 30% | **.8283** | .8262 | .5897 | .8889 | **.8901** | .4546 | **.7063** | .7005 | .5508 |
| 40% | **.8259** | .8255 | .4887 | .8620 | **.8787** | .2929 | **.7028** | .6968 | .4819 |
| 50% | **.8236** | .8225 | .3832 | .8542 | **.8678** | .2962 | **.7009** | .6814 | .4132 |

Table 4: Robust learning results with various noise levels on CIFAR-10, Flower, and HAM 10000 datasets. *Clean* is the performance of the predictor model when it is only trained with the samples with clean labels (e.g. at 20% noise level, it uses only 80% clean samples). *Baseline* is the performance of the predictor model when it is trained with both noisy and clean labels.

Noisy labels significantly degrade the prediction performance when they are included in the training dataset (see the increasing differences between *Baseline* and *Clean* in Table 4). DVRL demonstrates high robustness up to high noisy label ratio (50%). In some cases (even without noisy labels (i.e. 0% noise ratio)), the prediction performance even outperforms the *Clean* case, as DVRL prioritizes some clean samples more than others. Overall, DVRL framework is promising in maintaining high prediction performance even with a significant increase in the amount of noisy labels.

### C.2    ADDITIONAL RESULTS ON ROBUST LEARNING WITH NOISY FEATURES

In this section, we consider training with noisy input features, with a clean validation set. We independently add Gaussian noise with zero mean and a certain standard deviation of $\sigma$ to each feature in the training set independently. We use two tabular datasets (Adult and Blog) to evaluate the robustness of DVRL on input noise. As can be seen in Table 5, DVRL is robust with noise on the features and the performance gains are higher with larger noise in comparison to *Baseline* (i.e. treat all the noisy training samples equally), since DVRL can discover the training samples with less corrupted by the additive noise among the entire noisy training samples and provide higher weights on those less noisy samples.

| $\sigma$ | Blog | | Adult | |
|---|---|---|---|---|
| | *Baseline* | DVRL | *Baseline* | DVRL |
| 0.1 | 0.733 | **0.819** | 0.802 | **0.820** |
| 0.2 | 0.647 | **0.798** | 0.753 | **0.788** |
| 0.3 | 0.626 | **0.766** | 0.699 | **0.771** |
| 0.4 | 0.623 | **0.717** | 0.652 | **0.725** |

Table 5: Testing accuracy when trained with noisy features. $\sigma$ is the standard deviation of the added Gaussian noise, quantifying the level of perturbation on the features.

### C.3 ADDITIONAL RESULTS ON REMOVING HIGH/LOW VALUE SAMPLES WITH 20% LABEL NOISE

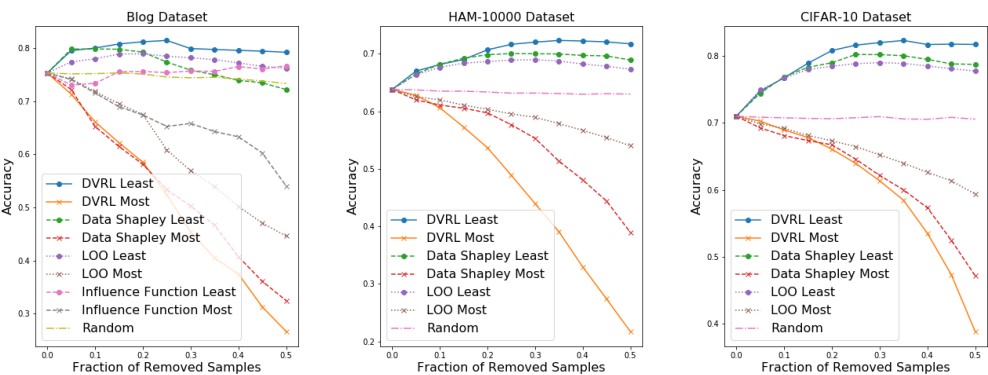

Figure 7: Prediction performance after removing the most and least important samples, according to the estimated data values. We assume a label noise with 20% ratio on (a) Blog, (b) HAM 10000, (c) CIFAR-10 datasets.

### C.4 ADDITIONAL RESULTS ON CORRUPTED SAMPLE DISCOVERY WITH 20% LABEL NOISE

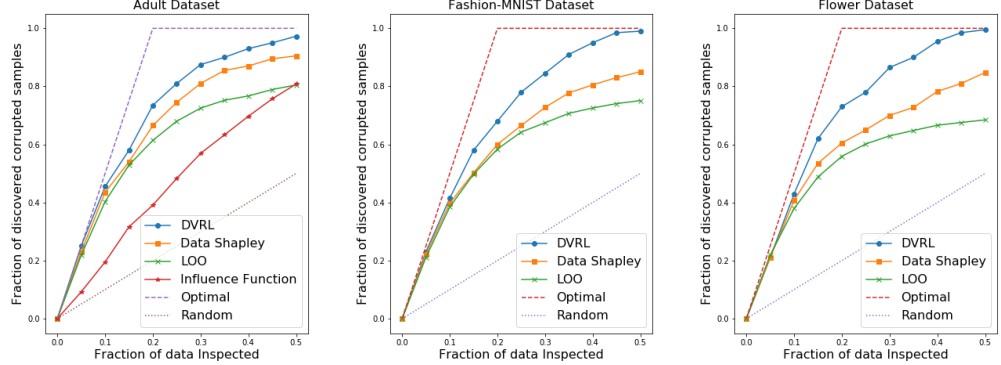

Figure 8: Discovering corrupted samples in three datasets ((a) Adult, (b) Fashion-MNIST, (c) Flower datasets) in the presence of 20% noisy labels. 'Optimal' saturates at the 20 % of the fraction, perfectly assigning the lowest data value scores to the samples with noisy labels. 'Random' does not introduce any knowledge on distinguishing clean vs. noisy labels, and thus the fraction of discovered corrupt samples is proportional to the amount of inspection.

### C.5 COMPARISON TO OTHER DOMAIN ADAPTATION BENCHMARKS

In this subsection, we compare DVRL to two established domain adaptation benchmarks: Adversarial Discriminative Domain Adaptation (ADDA) (Tzeng et al., 2017) and Domain Adversarial Neural Networks (DANN) (Ganin et al., 2016). We use the same experimental settings given in Table 3 using Rossmann Store Sales dataset with neural networks as the predictor model. Table 6 represents the domain adaptation results on 'Train on all' and 'Train on Rest' settings. As can be seen, DVRL yields superior (or similar in a few cases) compared to the two methods, ADDA and DANN, that are specifically designed for domain adaptation.

| Settings | Train on All | | | | Train on Rest | | | |
|---|---|---|---|---|---|---|---|---|
| Methods | Baseline | DVRL | ADDA | DANN | Baseline | DVRL | ADDA | DANN |
| A | 0.1531 | **0.1428** | 0.1465 | 0.1491 | 0.3124 | **0.2014** | 0.2119 | 0.2305 |
| B | 0.1529 | **0.0979** | 0.1193 | 0.1201 | 0.8071 | 0.5461 | **0.5444** | 0.5898 |
| C | 0.1620 | **0.1437** | 0.1503 | 0.1589 | 0.2153 | **0.1804** | 0.1871 | 0.1963 |
| D | 0.1459 | **0.1295** | 0.1351 | 0.1388 | 0.2625 | **0.1624** | 0.1910 | 0.2061 |

Table 6: Performance of *Baseline*, DVRL, ADDA, and DANN in train-on-all and train-on-rest settings with neural networks as the predictor model on the Rossmann Store Sales dataset. Metric is Root Mean Squared Percentage Error (RMSPE, lower the better). We use 79% of the data as training, 1% as validation, and 20% as testing.

### C.6 ABLATION STUDIES

In this subsection, we analyze the source of gains for three distinct components of DVRL: (1) discrete representations of data value estimator, (2) baseline for stabilizing the RL training, (3) output of the model trained on the clean validation set as the additional input (validation model). We report the corrupted sample discovery results where the experimental settings are same with Section 4.2.

| Models / Datasets | Blog | HAM-10000 | CIFAR-10 |
|---|---|---|---|
| DVRL | 47.3% | 60.2% | 68.1% |
| DVRL without sampler | 44.9% | 58.3% | 63.7% |
| DVRL without baseline | 45.8% | 56.6% | 62.9% |
| DVRL without validation model | 43.7% | 57.1% | 64.4% |
| Validation model only | 43.1% | 55.9% | 62.3% |

Table 7: Discovering corrupted samples in three datasets with 20% noisy label ratio. We report the fraction of discovered corrupted samples after inspecting 20% of the samples with multiple variants of DVRL (the higher the better).

As can be seen in Table 7, each component provides an additional gain in DVRL:

(1) A straightforward idea is to use the raw outputs of DVE to scale the contributions of each sample in the loss term, without using the sampler. Yet, we show the benefit of the discrete representation of DVE for data selection. The sampler encourages exploration of an extremely large action space in a systematic way. This helps DVE and predictor model to converge to a better optimal solution.

(2) Baseline stabilizes the convergence of reinforcement learning; thus, yields higher gains on complex datasets.

(3) The output of the validation model itself has informative signal as it achieves high performance (since it is trained with small-scale but high quality data). We observe that this signal helps DVRL, but even without this signal achieves high performance. We also observe that often a larger DVE model (with more iterations) is needed to estimate the data value in the absence of the informative signal from the validation model.

Note that we propose to use the output of the validation model as an additional input to the data valuation framework; thus, this can also be regarded as another contribution of our work. Also, the

output of the validation model is highly informative in the noisy sample discovery use case but not that significant in other applications such as domain adaptation or performance improvement by low value data removal in standard supervised learning setting.

# D  LEARNING CURVES OF DVRL

Fig. 9 shows the learning curves of DVRL on the noisy data (with 20% label noise) setting in comparison to the validation log loss without DVRL (directly trained on the noisy data without re-weighting) on 2 tabular datasets (Adult and Blog) and 4 image datasets (Fashion-MNIST, Flower, HAM 10000, and CIFAR-10).

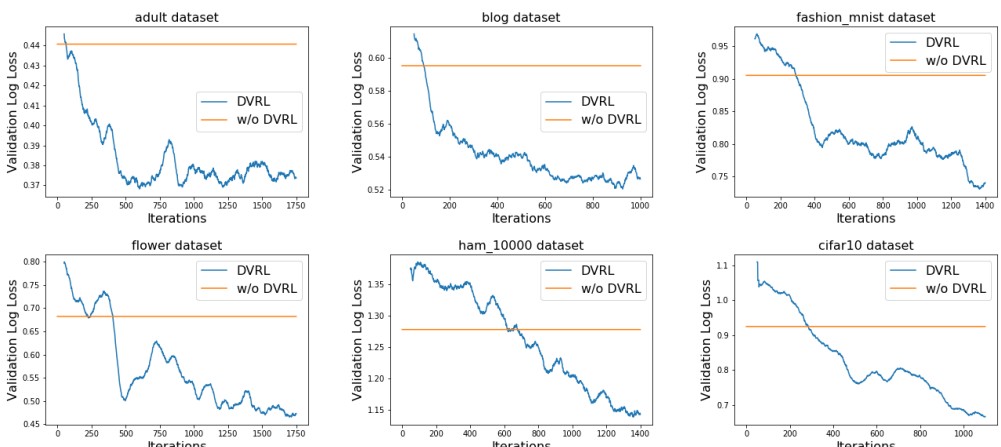

Figure 9: Learning curves of DVRL for 6 datasets with 20% noisy labels. x-axis: the number of iterations for data value estimator training, y-axis: validation performance (log loss). (Orange: validation log loss without DVRL, Blue: validation log loss with DVRL)

# E  CONFIDENCE INTERVALS OF DVRL PERFORMANCE ON CORRUPTED SAMPLE DISCOVERY EXPERIMENTS

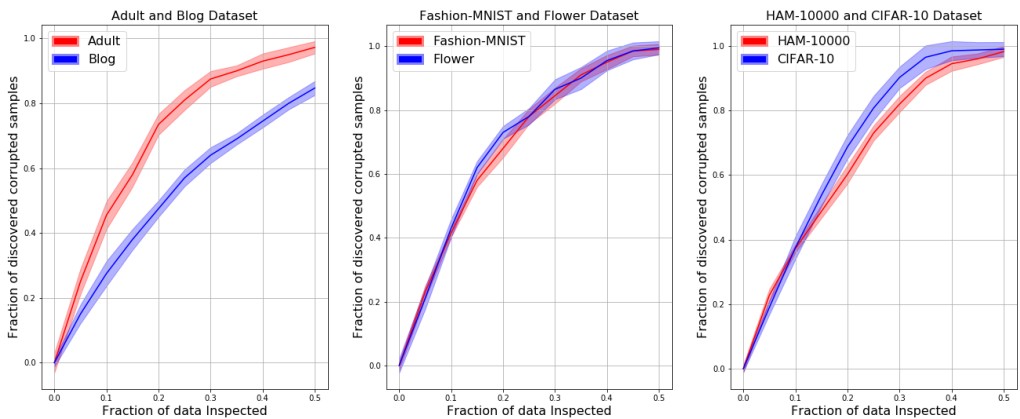

Figure 10: Corrupted sample discovery performance with 95% confidence intervals (computed by 10 independent runs) according to the estimated data values by DVRL. We assume a label noise with 20% ratio on (a) Adult and Blog, (b) Fashion-MNIST and Flower (c) HAM 10000 and CIFAR-10 datasets.

## F  ROSSMANN DATA STATISTICS & T-SNE ANALYSIS

| Store Type | A | B | C | D |
|---|---|---|---|---|
| # of Samples | 457042 (54.1%) | 15560 (1.8%) | 112968 (13.4%) | 258768 (30.6%) |
| Sales | 1390-1660-1854 | 2052-2459-2661 | 1753-1974-2178 | 2109-2355-2524 |
| Customers | 169-203-221 | 436-492-543 | 192-232-259 | 224-246-259 |

Table 8: Rossmann data statistics. Report 25-50-75 percentiles for sales and customers. # represents the number.

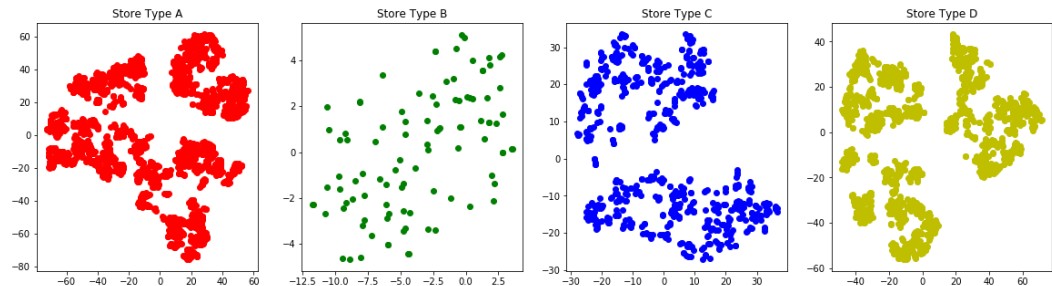

Figure 11: t-SNE analyses on the final layer representations of each store type in Rossmann dataset.

## G  FURTHER ANALYSIS ON ROSSMANN DATASET IN *Train on All* SETTING

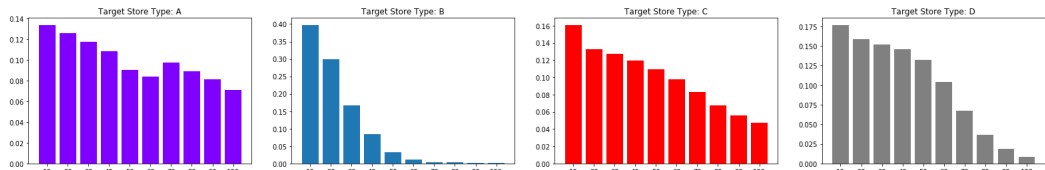

Figure 12: Histograms of the training samples from the target store type in *Train on All* setting based on the sorted data values estimated by DVRL. (x-axis: the sorted data values (in percentiles), y-axis: counts of training samples from the target store type (in ratio).

To further understand the results in *Train on All* setting, we sorted (in a decreasing order) the training samples by their data values estimated by DVRL and illustrate the distributions of the training samples that come from the target store type. As can be seen in Fig. 12, DVRL prioritizes the training samples which come from the same target store type.

