# OpenReview forum: "Data Valuation using Reinforcement Learning"
_ICLR.cc/2020/Conference — Reject_

### Official Review · AnonReviewer3 · 2019-10-21
**Official Blind Review #3**

**Rating:** 6

**Review:**

This paper proposes a meta learning approach based on data valuation for reinforcement learning tasks. The core idea is to train a second network (the data value estimator) in conjunction to a regular predictor network. The predictor is then trained with samples chosen via the data value estimation. The authors motivate this construction with the goal to filter out unreliable and corrupted data.

It's well established that RL poses a difficult learning problem, and as such the goal to improve the RL process is definitely a good one. To the best of my knowledge the approach proposed here is new. The exposition of the paper is also quite clear, and all parts of the approach are explained nicely. In addition, the submission contains a thorough evaluation of the method.

A central point for the method seems to be the validation data set which is used to train the data value estimator. The text emphasizes that this data set can be "small" several times, and the discussion and results of section 4.5 try to shed light here. However, Figure 5 indicates that a fairly large fraction of samples is needed to identify, e.g., more than 50% of the corrupted samples.

Another cricital aspect for meta-learning approaches such as this one is also the training time. RL is already expensive, so if the meta learning introduces a large factor, the training could quickly become infeasible. Here, the text gives a factor of about 3, which is noticeable, but not overly big. This still seems practical. Potentially, the estimator could also be reused (at least partially) for repeated training runs.

The tests in figure 2 are somewhat unituitive at first - I was expecting results of models trained on datasets with different samples being removed beforehand, rather than removing samples based on a trained estimator. However, this test makes sense on second sight, and e.g., the significant drop in performance after removing the most important samples indicates that the estimator was able to correctly identify a certain portion of data that is actually important for successful predictions. In addition to the noisy label and domain adaptation tests, this paints a positive picture. The method seems to yield useful (be it somewhat small) improvements in terms of learning performance.

One aspect that could be justified better in my opinion is the choise for a discrete representation for the data value estimator. Shouldn't the method likewise work with a continuous representation here? The paper explain how the discrete model is trained with quite some detail, but the choice itself could be motivated more clearly.

However, overall I think the method is interesting and yields nice performance improvements. It is described and evaluated in detail, so I think the paper could me included in the ICLR program.

**Experience Assessment:**

I have read many papers in this area.

**Review Assessment: Checking Correctness Of Derivations And Theory:**

I assessed the sensibility of the derivations and theory.

**Review Assessment: Checking Correctness Of Experiments:**

I assessed the sensibility of the experiments.

**Review Assessment: Thoroughness In Paper Reading:**

I read the paper thoroughly.

---

> ### Author Response · Authors · 2019-11-15
> **RE: Official Blind Review #3**
>
> We appreciate your positive comments and thank you for the various thoughtful comments that have helped us to improve the quality of our paper.  Please see our answers to the questions below.
>
> Answer 1: Thanks for pointing this out – we agree that this could be misunderstood.  In Figure 5, the x-axis represents the fraction of inspected data (not the validation data) and y-axis is the fraction of discovered corrupted samples. In this example, 20% of the samples are corrupted; thus, even in the optimal case, 10% of the samples are needed to be inspected to identify 50% of the corrupted samples. On Adult and Fashion-MNIST datasets, DVRL needs 13% and 14% of inspected samples to identify 50% of the corrupted samples respectively - merely 3% and 4% more than the optimal cases.  We have clarified this in the figure legend.
>
> Answer 2: We also feel that the overhead of DVRL compared to conventional training is practically feasible, and consider this one of the major benefits of our method compared to previous work.  We include a computational complexity analysis for DVRL in  Section 3 and Appendix B to provide detailed comparisons. Using the pre-trained model as the initialization of the predictor network to reduce the computational overhead of DVRL is a great idea and is actually what we did (see the end of Section 3).
>
> Answer 3: This is a great question.  We use the sampler in DVRL to obtain the discrete representation for the data values to encourage DVE model to efficiently explore the extremely large action (sample selection) space.  To quantitatively show the advantage of this design choice, we conducted an ablation study (see Appendix C.6 of the revised manuscript or the table below) with a variant of DVRL without using the sampler, and instead directly using the continuous outputs of the data value estimator.  DVRL with discrete representation of the data values outperforms DVRL without the sampler (continuous representation of the data values); for example, for corrupted sample discovery with the CIFAR-10 dataset, the performance gap is 4.4%.
>
> -------------------------------------------------------------------------------------------------
>              Models / Datasets             |   Blog   |   HAM-10000   |   CIFAR-10 |
> -------------------------------------------------------------------------------------------------
>                        DVRL                          |  47.3%  |       60.2%         |       68.1%  |
>           DVRL without sampler        |  44.9%  |       58.3%         |       63.7%   |
> -------------------------------------------------------------------------------------------------
>
> We hope that we have fully addressed your questions and concerns.  Please let us know if you have further comments.

---

### Official Review · AnonReviewer2 · 2019-10-23
**Official Blind Review #2**

**Rating:** 6

**Review:**

This paper proposes a method for assigning values to each datum.  For example, data with incorrect labels, data of low quality, or data from off-the-target distributions should be assigned low values. The main method involves training a neural network to predict the value for each training datum. The reward is based on performance on a small validation set. To make  gradient flow through data sampling, REINFORCE is used. The method is evaluated on multiple datasets. The results show that the proposed method outperforms a number of existing approaches.

I think the proposed method is reasonable, and the results look promising. However, I'm concerned that there's limited ablation study provided to show how each design choice impacts the performance. (After all, the proposed has many differences from existing methods.) Without proper ablation study, it's hard for the community to learn conclusively from the proposed techniques. In addition, as pointed out by the comments by Abubakar Abid, there is a model that is trained on the clean validation data used during training. But this is not discussed in paper. How does it impact performance? Also, all the image datasets studied in this paper are small, and this paper only considers fine-tuning the final layer from an ImageNet-pre-trained model. It'll be more convincing to show results on more relevant datasets or tasks in the community.

Overall I think this paper is slightly below the bar for publication in its current form, and will benefit from additional experiments.


-------after rebuttal--------
Thanks for providing additional results and explanations. I found the new ablations in C.6 helpful for understanding the impact of each of the design choices. The rebuttal also addresses my concerns regarding datasets, and my concerns regarding implementation details of ‘y_train_hat’, as now it's included in sec 4. Overall, after rebuttal, I'd like to recommend "weak accept" for this paper.

**Experience Assessment:**

I do not know much about this area.

**Review Assessment: Checking Correctness Of Derivations And Theory:**

N/A

**Review Assessment: Checking Correctness Of Experiments:**

I carefully checked the experiments.

**Review Assessment: Thoroughness In Paper Reading:**

I read the paper at least twice and used my best judgement in assessing the paper.

---

> ### Author Response · Authors · 2019-11-15
> **RE: Official Blind Review #2**
>
> Thank you for finding our work promising and providing constructive comments that have improved the quality of our paper.
>
> Answer 1: Thanks for this suggestion!  We have conducted additional ablation studies for three distinct cases and added to Appendix C.6 of the revised manuscript: (1) Discrete representations of data value estimator, (2) Baseline for stabilizing the RL training, (3) Output (y_train_hat) of the model trained on the clean validation set as the additional input (validation model). The table below shows the fraction of discovered corrupted samples (the same setting in Figure 4) after inspecting 20% of the samples with multiple variants of DVRL (higher is better).
>
> -------------------------------------------------------------------------------------------------
>              Models / Datasets             |   Blog   |   HAM-10000   |   CIFAR-10 |
> -------------------------------------------------------------------------------------------------
>                        DVRL                          |  47.3%  |       60.2%         |       68.1%  |
>           DVRL without sampler        |  44.9%  |       58.3%         |       63.7%   |
>           DVRL without baseline        |  45.8%  |       56.6%         |       62.9%   |
>         DVRL without y_train_hat     |  43.7%  |        57.1%        |       64.4%   |
>                y_train_hat only               |  43.1%  |        55.9%        |       62.3%   |
> -------------------------------------------------------------------------------------------------
>
> As can be seen from the above table, each distinct component provides an additional gain in DVRL.  Our intuitions for these improvements are as follows:
>  - The sampler (discrete representation of DVE for data selection) encourages exploration of an extremely large action space that helps DVRL to converge to the better optimal solution.
>  - Baseline stabilizes the convergence of RL training, thus yielding higher gains on complex datasets.
>  - The output of the validation model (y_train_hat) has an informative signal that helps DVRL, especially in the noisy sample discovery use case. Please see the next answer for further details.
>
> Answer 2: Thanks for this suggestion – we agree that it would be helpful to discuss that in the paper.  We have added the details of how we use the output (y_train_hat) of the model trained on the clean validation set to Section 4 of the revised paper (see “Experimental details”). ‘y_train_hat’ is defined as the difference between the predictions of a separate predictive model (fined-tuned or trained from scratch on the validation set) for the training samples and the original training labels. We use it as an additional input to the DVE model to further improve the performance of DVRL.
>
> To demonstrate the impact of this model empirically, we include an ablation study to show the performance comparison (1) using ‘y_train_hat’ only, (2) DVRL without using ‘y_train_hat’, (3) DVRL with using ‘y_train_hat’ in the Appendix C.6 of the revised manuscript.
>
> As can be seen in the above table (in Answer 1), y_train_hat has an informative signal to identify the noisy sample (y_train_hat alone achieves high performances).  DVRL without y_train_hat performs worse than DVRL but still achieves competitive performance. We also observe that we would need to use a larger DVE model (with more iterations) to estimate the data value in the absence of the informative signal y_train_hat.  y_train_hat is highly informative in the noisy sample discovery application but not that significant in other applications such as domain adaptation or performance improvement by low-value data removal in standard supervised learning setting.
>
> Answer 3: Thanks for this comment – we understand that selecting relevant and complex enough datasets and tasks is important.  The main reason that we included results on small-scale datasets and fine-tuning the final layer from an ImageNet-pretrained model is to compare DVRL with Data Shapley and LOO which are not scalable to large-scale datasets and complex models.  In addition to considering small datasets (results in Figures 2-4), we evaluated DVRL on two relatively large-scale image datasets (CIFAR-10 and CIFAR-100) that are commonly used in the academic community (see Table 1).  We also evaluated DVRL not only with fine-tuning the final layer from an ImageNet-pretrained model (in Figure 2-4) but also with more complex models when trained from scratch (ResNet-32, WideResNet-28-10) in Table 1. In addition we report results with a very large-scale tabular dataset (Rossmann) in Table 3 that contains 844k samples.  We feel that these results demonstrate well the promise for scalability and generality of DVRL for both large-scale datasets and complex models for image and other data types.  We hope this clarification is helpful.
>
> We hope that we have fully addressed your questions and concerns. Please let us know if you have further comments.

---

### Official Review · AnonReviewer1 · 2019-10-24
**Official Blind Review #1**

**Rating:** 3

**Review:**

This article present an approach to assign an importance value to an
observation, that quantifies its usefulness for a specific objective task.
This is useful in many contexts, such as domain adaptation, corrupted sample
discovery or robust learning.
The importance values may also be used to improve the performance of a model for the task.

The importance values are learned jointly with that model.
A small neural network called by the authors a Data Value Estimator (DVE) is
learnt by the authors to estimate sample selection probabilities, which will
dictate which instances will be used for the main model that tackles the
objective task.
While the main model is trained through usual mini-batch gradient descent, the DVE can
not be, since the sampling process is not differentiable.
It follows that the DVE is trained with a RL signal, that follows
the variation of the loss throughout the learning process.

The method proposed by the authors is new and show very significant results
over existing methods. It is scalable, while many of the presented approaches are not.
It is said to have a much lower computational burden than some existing methods,
e.g. LOO or Data Shapley.
The paper is very well written, and the method is illustrated on several datasets,
from different domains.

However, it seems that many approaches that did not suffer from the same complexity
drawbacks of LOO and Data Shapley were not compared to this work. While some
of the presented approaches are recent, e.g. ChoiceNet (2018), others are more established,
e.g. domain adversarial networks (DANs, Ganin et al 2016) and are not compared for
domain adaptation tasks. Given that the contributions of the authors are solely empirical,
it is necessary to compare their approach to other scalable domain adaptation approaches.
The approach proposed by the authors also features many hyperparameters, with
fixed chosen values, and the architecture of the DVE is not precised, which may impair the
reproducibility of the paper.

The authors should provide code if not already provided.
There is a mistake on the legends of Figure 2 and Figure 3, since accuracy
should increase when removing the least important samples.

**Experience Assessment:**

I have published one or two papers in this area.

**Review Assessment: Checking Correctness Of Derivations And Theory:**

I carefully checked the derivations and theory.

**Review Assessment: Checking Correctness Of Experiments:**

I carefully checked the experiments.

**Review Assessment: Thoroughness In Paper Reading:**

I read the paper thoroughly.

---

> ### Author Response · Authors · 2019-11-15
> **RE: Official Blind Review #1**
>
> We appreciate that you find our method is new, that our results are very significant, and that our manuscript is well-written. Thank you for your helpful suggestions that helped us to more strongly demonstrate the impact of our method for domain adaptation.  Please see below for answers to the questions as well as additional requested experiments.
>
> Answer 1: Thanks for the suggestion!  We would like to emphasize that domain adaptation is one of the many applications of data valuation framework and the mentioned established domain adaptation methods (e.g., DANN) cannot be generalized to other applications of data valuation in a straightforward way. Therefore, we mainly focused on Data Shapley and LOO as our main benchmarks to various data valuation applications in our manuscript. However, we do agree that it would be valuable to compare DVRL to other scalable domain adaptation methods. We have added Adversarial Discriminative Domain Adaptation (ADDA) and Domain Adversarial Neural Networks (DANN) as additional benchmarks in Appendix C.5 of the revised manuscript to compare with DVRL. The table below represents the domain adaptation results on ‘Train on all’ and ‘Train on Rest’ settings with neural network as the predictor model, similar to Table 3 in the manuscript.
>
> ----------------------------------------------------------------------------------------------------------------------------
>   Settings  |                         Train on all                        |                        Train on Rest                     |
> ----------------------------------------------------------------------------------------------------------------------------
>   Methods |  Baseline  |  DVRL  |  ADDA  |  DANN  |  Baseline  |  DVRL  |  ADDA  |  DANN  |
> ----------------------------------------------------------------------------------------------------------------------------
>         A         |     0.1531  | 0.1428 | 0.1465  |  0.1491  |    0.3124   |  0.2014  | 0.2119 | 0.2305 |
>         B         |     0.1529  | 0.0979 | 0.1193  |  0.1201  |    0.8071   |  0.5460  | 0.5444 | 0.5898 |
>         C         |     0.1620  | 0.1437 | 0.1503  |  0.1589  |    0.2153   |  0.1804  | 0.1871 | 0.1963 |
>         D         |     0.1459  | 0.1295 | 0.1351  |  0.1388  |    0.2625   |  0.1624  | 0.1910 | 0.2061 |
> ---------------------------------------------------------------------------------------------------------------------------
>
> As can be seen, DVRL yields superior (or similar in a few cases) compared to the two methods, ADDA and DANN, that are specifically designed for domain adaptation.
>
> In addition, for robust learning with noisy samples (Section 4.3), we already compare with other scalable models such as Learning to Reweight and MentorNet in Table 1.
>
> Answer 2: We agree that reproducibility is very important. In the experimental details subsection (at the beginning of Section 4), we had the details of the experimental hyper-parameters that are important for reproducing the reported results. We have added further details to this Section which should help clarifying architecture-related ambiguities:
> “As the DVE architecture, for tabular datasets, we use 5-layer perceptrons with 100 hidden units and ReLU; and for image datasets, we use 5-layer perceptrons with 100 hidden units and ReLU on top of the CNN-based architecture used for the predictor network (such as ResNet-32 or WideResNet-28-10 in Table 1) as DVE architecture.”
>
> In addition, as answered below, we also published the source codes when we submitted the manuscript (9/25/2019) via OpenReview website which should ensure reproducibility.
>
> Answer 3: We published the code when we submitted the manuscript (9/25/2019) via the OpenReview website.
>
> Answer 4: Thank you for pointing this typo out. We have fixed the typo as follows: “Prediction performance after removing the most (marked with cross) and least (marked with circle) important samples” - please see the revised manuscript.
>
> We hope that we have fully addressed your reproducibility and comparison concerns. Please let us know if you have further comments.

---

### Public Comment · ~Abubakar_Abid1 · 2019-10-21
**Reproducibility Concerns**

I have carefully tried to reproduce the basic experiment to detect corrupted labels on the Adult Dataset, and I have the following concerns:

(1) The code that is provided on GitHub includes side information into the data valuation estimator (https://git.io/JeRRR ). This side information comes from a model that is *trained on the clean validation data* and used to make predictions on each batch of training data. The difference between the label of the training data and prediction is fed into the data value estimator. In the comments in the code, the authors remark that "adding y_hat_input into data value estimator as the additional input is observed to improve data value estimation quality in some cases," but this is a significant piece of side information and I don't see a discussion of this in the paper. Ideally, there should be an ablation experiment that quantifies the contribution of this side information.

(2) In fact, in my own preliminary experiments on noisy label recovery in the Adult dataset, I believe this side information may be providing the bulk of the information used to train the data value estimation network. Removing this information drops the performance significantly, while the performance remains almost unchanged with all of the other inputs to the data value estimator are removed. If this is the case, then there is a much simpler method to be proposed: a method that simply uses the prediction differences from a model trained clean validation data to quantify the relative value of training data, or at least to identify corrupted labels in the training data.

---

> ### Author Response · Authors · 2019-10-25
> **Answer to the public comment**
>
> Thank you for your interest in our paper. Below are the answers to your questions:
>
> (1) In order to provide informative signal to data valuator, we propose to use ‘y_train_hat’ as an additional input to the data valuator. ‘y_train_hat’ is defined as the difference between the predictions of a separate predictive model (fined-tuned or trained from scratch on the validation set) for the training samples and the original training labels. Note that ‘y_train_hat’ is obtained from the ‘given’ training and validation sets so it is a fair side information. Extra side information in ‘y_train_hat’ is noticeably valuable only in some cases. Intuitively, if the training label is corrupted, ‘y_train_hat’ would be high; thus, this could be an important signal for the data valuator to assign low value to this sample. Besides, in domain adaptation setting, if the label distribution is different between different domains, ‘y_train_hat’ could also be somewhat important signal for the data valuator to distinguish samples from different distributions. We omitted to explain this in the original manuscript due to the page limitation; however, we will clarify this in the revised manuscript.
>
> (2) The information in ‘y_train_hat’ could be important in some cases, but the actual challenge and our major contribution is proposing a method to optimally utilize such feedback signal in end-to-end training. For example, straightforward methods like directly using ‘y_train_hat’ as the values of samples would yield limited performance across various applications from our experiments. With DVRL framework, this signal is properly utilized and used towards high performance data valuation and robust learning.
>
> Even without using ‘y_train_hat’ as the additional input to the data valuator, DVRL still yields high quality data valuation results. In some cases, hyper-parameters (e.g. learning rate or the data valuator model complexity) need to be re-optimized but the ultimate achievable performance should not be very different.
>
> The main benefit of including ‘y_train_hat’ signal (proposed in this paper) is faster convergence in some cases. To demonstrate this empirically, we will include an ablation study for performance comparison (1) using ‘y_train_hat’ only, (2) DVRL without using ‘y_train_hat’, (3) DVRL with using ‘y_train_hat’ in the Appendix of the revised manuscript.

---

### Decision · Program_Chairs · 2019-12-19

**Decision:**

Reject

**Comment:**

The paper suggests an RL-based approach to design a data valuation estimator. The reviewers agree that the proposed method is new and promising, but they also raised concerns about the empirical evaluations, including not comparing with other approaches of data valuation and limited ablation study.
The authors provided a rebuttal to address these concerns. It improves the evaluation of one of the reviewers, but it is difficult to recommend acceptance given that we did not have a champion for this paper and the overall score is not high enough.